# Physics-Informed Deep B-Spline Networks

## Abstract

Physics-informed machine learning provides an approach to combining data and governing physics laws for solving complex partial differential equations (PDEs). However, efficiently solving PDEs with varying parameters and changing initial conditions and boundary conditions (ICBCs) remains an open challenge. We propose a hybrid framework that uses a neural network to learn B-spline control points to approximate solutions to PDEs with varying system and ICBC parameters. The proposed network can be trained efficiently as one can directly specify ICBCs without imposing losses, calculate physics-informed loss functions through analytical formulas, and requires only learning the weights of B-spline functions as opposed to both weights and basis as in traditional neural operator learning methods. We show theoretical guarantees that the proposed B-spline networks are universal approximators of arbitrary dimensional PDEs under certain conditions. We also demonstrate in experiments that the proposed B-spline network can solve problems with discontinuous ICBCs and outperforms existing methods, and is able to learn solutions of 3D heat equations with diverse initial conditions.

## 1 Introduction

Recent advances in scientific machine learning have boosted the development for solving complex partial differential equations (PDEs). Physics-informed neural networks (PINNs) are proposed to combine information of available data and the governing physics model to learn the solutions of PDEs (Raissi et al., 2019; Han et al., 2018). However, in the real world the parameters for the PDE and for the initial and boundary conditions can be changing, and solving PDEs for all possible parameters can be important but demanding. For example in a safety-critical control scenario, the system dynamics and the safe region can vary over time, resulting in changing parameters for the PDE that characterizes the probability of safety. On the other hand, solving such PDEs is important for safe control but can be hard to achieve in real time with limited online computation. In general, to account for parameterized PDEs and varying initial conditions and boundary conditions (ICBCs) in PINNs is challenging, as the solution space becomes much larger (Karniadakis et al., 2021). To tackle this challenge, parameterized PINNs are proposed (Cho et al., 2024). Plus, a new line of research on neural operators is conducted to learn operations of functions instead of the value of one specific function (Kovachki et al., 2023; Li et al., 2020; Lu et al., 2019). Nevertheless, such methods can not efficiently handle problems with irregular initial and boundary conditions.

In this work, we leverage the advantages of B-spline functions and physics-informed learning, to form physics-informed deep B-spline networks (PI-DBSN) to efficiently learn parameterized PDEs with varying initial and boundary conditions (Fig. 1). The network composites of B-spline basis functions, and a parameterized neural network that learns the weights for the B-spline basis. Specifically, the coefficient network takes inputs of the PDE and ICBC parameters, and outputs the control points tensor (*i.e.,* weights of B-splines). Then this control points tensor is multiplied with the B-spline basis to produce the final output as the approximation of PDEs. One can evaluate the prediction of the PDE solution at any point, and we use physics loss and data loss to train the network similar to PINNs (Cuomo et al., 2022). There are several advantages for the proposed PI-DBSN:

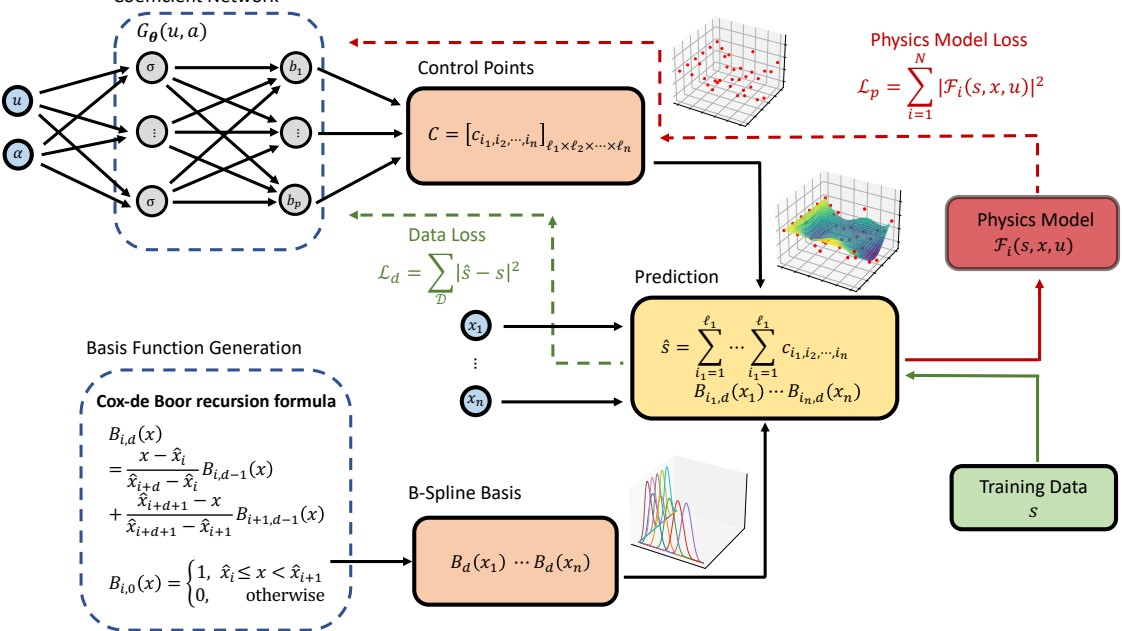

Figure 1: Diagram of PI-DBSN. The coefficient network takes system and ICBC parameters as input and outputs the control points tensor, which is then multiplied with the B-spline basis to produce the final output. Physics and data losses are imposed to train the network. Solid lines depict the forward pass, and dashed lines depict the backward pass of the network.

1. The B-spline basis functions are fixed and can be pre-calculated before training, thus we only need to train the coefficient network which saves computation and stabilizes training.

2. The B-spline functions have analytical expressions for its gradients and higher-order derivatives, which provide faster and more accurate calculation for the physics-informed losses during training over automatic differentiation.

3. Due to the properties of B-splines, we can directly specify Dirichlet boundary conditions and initial conditions through the control points tensor without imposing loss functions, which helps with learning extreme and complex ICBCs.

The rest of the paper is organized as follow. We discuss related work in Sec. 2, and introduce our proposed PI-DBSN in Sec. 3. We then show in Sec. 4 that despite the use of fixed B-spline basis, the PI-DBSN is a universal approximator and can learn high-dimensional PDEs. Following the theoretical analysis, in Sec. 5 we demonstrate with experiments that PI-DBSN can solve problems with discontinuous ICBCs and outperforms existing methods, and is able to learn high-dimensional PDEs. Finally, we conclude the paper in Sec. 6.

## 2 RELATED WORK

**PINNs:** Physics-informed neural networks (PINNs) are neural networks that are trained to solve supervised learning tasks while respecting any given laws of physics described by general nonlinear partial differential

equations (Raissi et al., 2019; Han et al., 2018; Cuomo et al., 2022). PINNs take both data and the physics model of the system into account, and are able to solve the forward problem of getting PDE solutions, and the inverse problem of discovering underlying governing PDEs from data. PINNs have been widely used in power systems (Misyris et al., 2020), fluid mechanics (Cai et al., 2022) and medical care (Sahli Costabal et al., 2020), *etc*. Different variants of PINN have been proposed to meet different design criteria, for example Bayesian physics-informed neural networks are used for forward and inverse PDE problems with noisy data (Yang et al., 2021), physics-informed neural networks with hard constraints are proposed to solve topology optimizations (Lu et al., 2021b), and parallel physics-informed neural networks via domain decomposition are proposed to solve multi-scale and multi-physics problems (Shukla et al., 2021). It is shown that under certain assumptions that PINNs have bounded error and converge to the ground truth solutions (De Ryck & Mishra, 2022; Mishra & Molinaro, 2023; 2022; Fang, 2021; Pang et al., 2019; Jiao et al., 2021). We leverage the idea of physics-informed learning as we constrain the network output to satisfy physics laws, but use a novel B-spline formulation for more efficient training for families of PDEs.

**Neural Operators:** Neural operators are a class of deep learning architectures designed to learn maps between infinite-dimensional function spaces instead of values of specific functions (Kovachki et al., 2023). DeepONets (Lu et al., 2019; 2021a) and Fourier Neural Operators (FNOs) (Li et al., 2020) are two common approaches along this line of research, and a detailed comparison can be found in Lu et al. (2022). In Wang et al. (2021) DeepONets are combined with physics-informed learning to solve fixed PDEs. Generalizations of DeepONet (Gao et al., 2021) and FNO (Li et al., 2024) consider learning (state) parameterized PDEs with fast evaluation, but training is usually slow. Recent work (Kumar et al., 2024) incorporates multi-task mechanism within DeepONet to learn PDEs with varying conditions, but unique and manually designed polynomial representation of the varying parameter is needed as input to the branch net of the system. While all DeepONet-based methods need to train two networks at a time (branch and trunk net in the architecture) and impose losses on ICBCs, our method directly specifies ICBCs, and uses fixed B-spline functions as the basis such that only one coefficient network is trained for better efficiency and stability.

**B-splines + NN:** B-splines are piece-wise polynomial functions derived from slight adjustments of Bezier curves, aimed at obtaining polynomial curves that tie together smoothly (Ahlberg et al., 2016). B-splines have been widely used in signal and imaging processing (Unser, 1999; Lehmann et al., 2001), computer aided design (Riesenfeld, 1973; Li, 2020), *etc*. B-splines are also used to assist in solving PDEs, for example Jia et al. (2013) uses B-spline in finite element methods for PDE solving, and in Song et al. (2022) spline-inspired mesh movement networks are proposed to solve PDEs. B-splines together with neural networks (NNs) are used for surface reconstruction (Iglesias et al., 2004), nonlinear system modeling (Yiu et al., 2001; Wang et al., 2022b), and controller design for dynamical systems (Chen et al., 2004; Deng et al., 2008). In Fakhoury et al. (2022) and Doległo et al. (2022) NNs are used to learn weights for B-spline functions to approximate fixed ODEs and PDEs, respectively. Note that the recently proposed Kolmogorov–Arnold Networks (KANs) also uses splines in neural networks (Liu et al., 2024), but is different from our work. In KAN the spline functions are used to produce learnable weights in the NN as an alternative architecture to multi-layer perceptrons (MLPs). In our work the B-splines are fixed and the neural network can take arbitrary MLP/non-MLP based architectures including KANs.

## 3 PROPOSED METHOD

### 3.1 PROBLEM FORMULATION

The goal of this paper is to efficiently estimate high-dimensional surfaces with corresponding governing physics laws of a wide range variety of parameters (*e.g.,* the solution of a family of ODEs/PDEs). We denote $s : \mathbb{R}^n \to \mathbb{R}$ as the ground truth, *i.e.,* $s(x)$ is the value of the surface at point $x$, where $x \in \mathbb{R}^n$. We assume

the physics laws can be written as

$$\mathcal{F}_i(s, x, u) = 0, \ x \in \Omega_i(\alpha), \ \forall i = 1, \cdots, N, \tag{1}$$

where $u \in \mathbb{R}^m$ is the parameters of the physics systems, $N$ is the number of governing equations, $\Omega_i(\alpha) \in \mathbb{R}^n$ parameterized by $\alpha$ is the region that the $i$-th physics law applies. We denote $\Omega \in \mathbb{R}^n$ the general region of interest, and in this paper we consider $n$-dimensional bounded domain $\Omega = [a_1, b_1] \times [a_2, b_2] \times \cdots \times [a_n, b_n]$. Our goal is to generate $\hat{s}$ with neural networks to estimate $s$ on the entire domain of $\Omega$, with all possible parameters $u$ and $\alpha$. For example, in the case of solving 2D heat equations on $(x_1, x_2) \in [0, \alpha]^2$ at time $t \in [0, 10]$ with varying coefficient $u \in [0, 2]$ and $\alpha \in [3, 4]$, we have the physics laws to be

$$\mathcal{F}_1(s, x, u) = \partial s / \partial t - u \left( \partial^2 s / \partial x_1^2 + \partial^2 s / \partial x_2^2 \right) = 0, \qquad x = (x_1, x_2, t) \in \Omega_x \times \Omega_t, \tag{2}$$

$$\mathcal{F}_2(s, x, u) = s - 1 = 0, \qquad x = (x_1, x_2, t) \in \partial\Omega_x \times \Omega_t \tag{3}$$

where $\Omega_x = [0, \alpha]^2$ and $\Omega_t = [0, 10]$, and $\partial\Omega_x$ is the boundary of $\Omega_x$. Here, equation 2 is the heat equation and equation 3 is the boundary condition. In this case, we want to solve for $s$ on $\Omega = \Omega_x \times \Omega_t$ for all $u \in [0, 2]$ and $\alpha \in [3, 4]$. Similar problems have been studied in (Li et al., 2024; Gao et al., 2021; Cho et al., 2024) while the majority of the literature considers solving parameterized PDEs but with either fixed coefficients or fixed domain and initial/boundary conditions. We slightly generalize the problem to consider systems with varying parameters, and with potential varying domains and initial/boundary conditions.

## 3.2 B-SPLINES WITH BASIS FUNCTIONS

In this section, we introduce one-dimensional B-splines. For state space $x \in \mathbb{R}$, the B-spline basis functions are given by the Cox-de Boor recursion formula:

$$B_{i,d}(x) = \frac{x - \hat{x}_i}{\hat{x}_{i+d} - \hat{x}_i} B_{i,d-1}(x) + \frac{\hat{x}_{i+d+1} - x}{\hat{x}_{i+d+1} - \hat{x}_{i+1}} B_{i+1,d-1}(x), \tag{4}$$

and

$$B_{i,0}(x) = \begin{cases} 1, & \hat{x}_i \leq x < \hat{x}_{i+1}, \\ 0, & \text{otherwise.} \end{cases} \tag{5}$$

Here, $B_{i,d}(x)$ denotes the value of the $i$-th B-spline basis of order $d$ evaluated at $x$, and $\hat{x}_i \in (\hat{x}_i)_{i=1}^{\ell+d+1}$ is a non-decreasing vector of knot points. Since a B-spline is a piece-wise polynomial function, the knot points determine in which polynomial the parameter $x$ belongs. While there are multiple ways of choosing knot points, we use $(\hat{x}_i)_{i=1}^{\ell+d+1}$ with $\hat{x}_1 = \hat{x}_2 = \cdots = \hat{x}_{d+1}$ and $\hat{x}_{\ell+1} = \hat{x}_{\ell+2} = \cdots = \hat{x}_{\ell+d+1}$, and for the remaining knot points we select equispaced values. For example on $[0, 3]$ with number of control points $\ell = 6$ and order $d = 3$, we have $\hat{x} = [0, 0, 0, 0, 1, 2, 3, 3, 3, 3]$, in total $\ell + d + 1 = 10$ knot points.

We then define the control points

$$c := [c_1, c_2, \ldots, c_\ell], \tag{6}$$

and the B-spline basis functions vector

$$B_d(x) := [B_{1,d}(x), B_{2,d}(x), \ldots, B_{\ell,d}(x)]^\top. \tag{7}$$

Then, we can approximate a solution $s(x)$ with

$$\hat{s}(x) = c B_d(x). \tag{8}$$

Note that with our choice of knot points, we ensure the initial and final values of $\hat{s}(x)$ coincide with the initial and final control points $c_1$ and $c_\ell$. This property will be used later to directly impose initial conditions and Dirichlet boundary conditions with PI-DBSN.

### 3.3 MULTI-DIMENSIONAL B-SPLINES

Now we extend the B-spline scheme to the multi-dimensional case. We start by considering the 2D case where $x = [x_1, x_2]^\top \in \mathbb{R}^2$. Along each dimension $x_i$, we can generate B-spline basis functions based on the Cox-de Boor recursion formula in equation 4 and equation 5. We denote the B-spline basis of order $d$ as $B_{i,d}(x_1)$, $B_{j,d}(x_2)$ for the $i$-th and $j$-th function of $x_1$ and $x_2$, respectively. Then with a control points matrix $C = [c_{i,j}]_{\ell \times p}$, the 2-dimensional surface can be approximated by the B-splines as

$$s(x_1, x_2) \approx \sum_{i=1}^{\ell} \sum_{j=1}^{p} c_{i,j} B_{i,d}(x_1) B_{j,d}(x_2), \tag{9}$$

where $\ell$ and $p$ are the number of control points along the 2 dimensions. This can be written in the matrix multiplication form as

$$\hat{s}(x_1, x_2) = B_d(x_1)^\top C B_d(x_2) = [B_{1,d}(x_1), \cdots, B_{1,\ell}(x_1)] \begin{bmatrix} c_{1,1} & \cdots & c_{1,p} \\ \vdots & \vdots & \vdots \\ c_{\ell,1} & \cdots & c_{\ell,p} \end{bmatrix} \begin{bmatrix} B_{1,d}(x_2) \\ \vdots \\ B_{p,d}(x_2) \end{bmatrix}, \tag{10}$$

where $\hat{s}(x_1, x_2)$ is the approximation of the 2D solution at $(x_1, x_2)$, $C$ is the control points matrix and $B_d(x_1)$ and $B_d(x_2)$ are the B-spline vectors defined in equation 7.

More generally, for a $n$-dimensional space $x = [x_1, \cdots, x_n] \in \mathbb{R}^n$, we can generate B-spline basis functions based on the Cox-de Boor recursion formula along each dimension $x_i$ with order $d_i$ for $i = 1, 2, \cdots, n$, and the $n$-dimensional control point tensor will be given by $C = [c_{i_1, i_2, \cdots, i_n}]_{\ell_1 \times \ell_2 \times \cdots \times \ell_n}$, where $i_k$ is the $k$-th index of the control point, and $\ell_k$ is the number of control points along the $k$-th dimension. We can then approximate the $n$-dimensional surface with B-splines and control points via

$$\hat{s}(x_1, x_2, \cdots, x_n) = \sum_{i_1=1}^{\ell_1} \sum_{i_2=1}^{\ell_2} \cdots \sum_{i_n=1}^{\ell_n} c_{i_1, i_2, \cdots, i_n} B_{i_1, d_1}(x_1) B_{i_2, d_2}(x_2) \cdots B_{i_n, d_n}(x_n). \tag{11}$$

### 3.4 PHYSICS-INFORMED B-SPLINE NETS

In this section, we introduce our proposed physics-informed deep B-spline networks (PI-DBSN). The overall diagram of the network is shown in Fig. 1. The network composites a coefficient network that learns the control point tensor $C$ with system parameters $u$ and ICBC parameters $\alpha$, and the B-spline basis functions $B_{d_i}$ of order $d_i$ for $i = 1, \cdots, n$. During the forward pass, the control point tensor $C$ output from the coefficient net is multiplied with the B-spline basis functions $B_{d_i}$ via equation 11 to get the approximation $\hat{s}$. For the backward pass, two losses are imposed to efficiently and effectively train PI-DBSN. We first impose a physics model loss $\mathcal{L}_p = \sum_{i=1}^{N} \sum_{x \in \mathcal{P}} \frac{1}{|\mathcal{P}|} |\mathcal{F}_i(s, x, u)|^2$ where $\mathcal{F}_i$ is the governing physics model of the system as defined in equation 1, and $\mathcal{P}$ is the set of points sampled to evaluated the governing physics model. When data is available, we can additionally impose a data loss $\mathcal{L}_d = \frac{1}{|\mathcal{D}|} \sum_{x \in \mathcal{D}} |s(x) - \hat{s}(x)|^2$ to capture the mean square error of the approximation, where $s$ is the data point for the high dimensional surface, $\mathcal{D}$ is the data set, and $\hat{s}$ is the prediction from the PI-DBSN. The total loss is given by $\mathcal{L} = w_p \mathcal{L}_p + w_d \mathcal{L}_d$ where $w_p$ and $w_d$ are the weights for physics and data losses, and are usually set to values close to 1. We use $G_{\boldsymbol{\theta}}(u, \alpha)(x)$ to denote the PI-DBSN parameterized by $\boldsymbol{\theta}$, where $(u, \alpha)$ is the input to the coefficient net (parameters of the system and ICBCs), and $x$ will be the input to the PI-DBSN (the state and time in PDEs). With this notation we have $C = G_{\boldsymbol{\theta}}(u, \alpha)$ and $\hat{s}(x) = G_{\boldsymbol{\theta}}(u, \alpha)(x)$.

Note that several good properties of B-splines are leveraged in PI-DBSN.

**First, the derivatives of the B-spline functions can be analytically calculated.** Specifically, the $p$-th derivative of the $d$-th ordered B-spline is given by (Butterfield, 1976)

$$\frac{d^p}{dx^p} B_{i,d}(x) = \frac{(d-1)!}{(d-p-1)!} \sum_{k=0}^{p} (-1)^k \binom{p}{k} \frac{B_{i+k,d-p}(x)}{\prod_{j=0}^{p-1} (\hat{x}_{i+d-j-1} - \hat{x}_{i+k})}. \tag{12}$$

Given this, we can directly calculate these values for the back-propagation of physics model loss $\mathcal{L}_p$, which improves both computation efficiency and accuracy over automatic differentiation that is commonly used in physic-informed learning (Cuomo et al., 2022).

**Besides, any Dirichlet boundary conditions and initial conditions can be directly assigned via the control points tensor without any learning involved.** This is due to the fact that the approximated solution $\hat{s}$ at the end points along each axis will have the exact value of the control point. For example, in a 2D case when the initial condition is given by $s(x,0) = 0, \forall x$, we can set the first column of the control points tensor $c_{i_1,1} = 0$ for all $i_1 = 1, \cdots, \ell_1$ and this will ensure the initial condition is met for the PI-DBSN output. This greatly enhances the accuracy of the learned solution near initial and boundary conditions, and improves the ease of design for the loss function as weight factors are often used to impose stronger initial and boundary condition constraints in previous literature (Wang et al., 2022a). We will demonstrate later in the experiment section where we compare the proposed PI-DBSN with physic-informed DeepONet that this feature will result in better estimation of the PDEs when the initial and boundary conditions are hard to learn.

**Furthermore, better training stability can be obtained**. The B-spline basis functions are fixed and can be calculated in advance, and training is involved only for the coefficient net.

## 4 THEORETICAL ANALYSIS

In this section, we provide theoretical guarantees of the proposed PI-DBSN on learning high-dimensional PDEs. We first show that B-splines are universal approximators, and then show that with combination of B-splines and neural networks, the proposed PI-DBSN is a universal approximator under certain conditions. At last we argue that when the physics loss is densely imposed and the loss functions are minimized, the network can learn unique PDE solutions. All theorem proofs can be found the in the Appendix of the paper.

We first consider the one-dimensional function space $L_2([a,b])$ with $L_2$ norm defined over the interval $[a,b]$. For two functions $s, g \in L_2([a,b])$, we define the inner product of these two functions as

$$\langle s, g \rangle := \int_a^b s(x)g^*(x)dx, \tag{13}$$

where $*$ denotes the conjugate complex. We say a function $s(x)$ is square-integrable if the following holds

$$\langle s, s \rangle = \int_a^b |s(x)|^2 dx < \infty. \tag{14}$$

We define the $L_2$ norm between two functions $s, g$ as

$$\|s - g\|_2 := \left( \int_a^b |s(x) - g(x)|^2 dx \right)^{\frac{1}{2}}. \tag{15}$$

We then state the following theorem that shows B-spline functions are universal approximators in the sense of $L_2$ norms in one dimension.

**Theorem 1.** *Given a positive natural number $d$ and any $d$-time differentiable function $s(x) \in L_2([a,b])$, then for any $\epsilon > 0$, there exist a positive natural value $\ell$, and a realization of control points $c_1, c_2, \cdots, c_\ell$ such that*

$$\|s - \hat{s}\|_2 \leq \epsilon, \tag{16}$$

*where*

$$\hat{s}(x) = \sum_{i=1}^{\ell} c_i B_{i,d}(x)$$

*is the B-spline approximation with $B_{i,d}(x)$ being the B-spline basis functions defined in equation 7.*

Now that we have the error bound of B-spline approximations in one dimension, we will extend the results to arbitrary dimensions. We point out that the space $L_2([a, b])$ is a Hilbert space (Balakrishnan, 2012). Let us consider $n$ Hilbert spaces $L_2([a_i, b_i])$ for $i = 1, 2, \cdots, n$. We define the inner products of two $n$-dimensional functions $s, g \in L_2([a_1, b_1] \times \cdots \times [a_n, b_n])$ as

$$\langle s, g \rangle := \int_{a_n}^{b_n} \cdots \int_{a_1}^{b_1} s(x_1, \cdots, x_n) g^*(x_1, \cdots, x_n) dx_1 \cdots dx_n, \tag{17}$$

and we say a function $s : \mathbb{R}^n \to \mathbb{R}$ is square-integrable if

$$\langle s, s \rangle = \int_{a_n}^{b_n} \cdots \int_{a_1}^{b_1} |s(x_1, \cdots, x_n)|^2 dx_1 \cdots dx_n < \infty. \tag{18}$$

Now we present the following lemma to bound the approximation error of $n$-dimensional B-splines.

**Lemma 2.** *Given a set positive natural numbers $d_1, \cdots, d_n$ and a $d$-time differentiable function $s(x_1, x_2, \cdots, x_n) \in L_2([a_1, b_1] \times [a_2, b_2] \times \cdots \times [a_n, b_n])$. Assume $d \geq \max\{d_1, \cdots, d_n\}$, then given any $\epsilon > 0$, there exist $\ell_i \in \mathbb{N}^+$ of control points for each component $i = 1, ..., n$, such that*

$$\|s(x_1, x_2, \cdots, x_n) - \hat{s}(x_1, x_2, \cdots, x_n)\|_2 \leq \epsilon, \tag{19}$$

*where*

$$\hat{s}(x_1, x_2, \cdots, x_n) = \sum_{i_1=1}^{\ell_1} \sum_{i_2=1}^{\ell_2} \cdots \sum_{i_n=1}^{\ell_n} c_{i_1, i_2, \cdots, i_n} B_{i_1, d_1}(x_1) B_{i_2, d_2}(x_2) \cdots B_{i_n, d_n}(x_n). \tag{20}$$

On the other hand, we know that neural networks are universal approximators (Hornik et al., 1989; Leshno et al., 1993), *i.e.,* with large enough width or depth a neural network can approximate any function with arbitrary precision. We restate the universal approximation theorem in our context assuming the requirements for the neural network are met. [1]

**Theorem 3.** *Given any $u$ and $\alpha$ in a finite parameter set, and any control points tensor $C := [c]_{\ell_1 \times \cdots \times \ell_n}$, for the coefficient net $G_{\boldsymbol{\theta}}(u, \alpha)$ and $\forall \epsilon > 0$, when the network has enough width and depth, there is $\boldsymbol{\theta}^*$ such that*

$$\|G_{\boldsymbol{\theta}^*}(u, \alpha) - C\| \leq \epsilon. \tag{21}$$

Then, we combine Lemma 2 and Theorem 3 to show the universal approximation property of PI-DBSN.

**Theorem 4.** *For any $n \in \mathbb{N}^+$ dimension, any $u$ and $\alpha$ in a finite parameter set, let $d_i$ be the order of B-spline basis for dimension $i = 1, 2, \cdots, n$. Then for any $d$-time differentiable function $s(x_1, x_2, \cdots, x_n) \in L_2([a_1, b_1] \times [a_2, b_2] \times \cdots \times [a_n, b_n])$ with $d \geq \max\{d_1, \cdots, d_n\}$ where the domain depends on $\alpha$ and the function depends on $u$, and any $\epsilon > 0$, there exist a PI-DBSN configuration $G_{\boldsymbol{\theta}}(u, \alpha)$ with enough width and depth, and corresponding parameters $\boldsymbol{\theta}^*$ independent of $u$ and $\alpha$ such that*

$$\|\tilde{s} - s\|_2 \leq \epsilon, \tag{22}$$

*where $\tilde{s} = G_{\boldsymbol{\theta}^*}(u, \alpha)(x)$ is the B-spline approximation defined in equation 11 with the control points tensor $G_{\boldsymbol{\theta}^*}(u, \alpha)$.*

---

[1]The Borel space assumptions are met since we consider $L_2$ space which is a Borel space.

Theorem 4 tells us the proposed PI-BDSN is an universal appproximator of high-dimensional surfaces with varying parameters and domains. Thus we know that when the solution of the problem defined in equation 1 is unique, and the physics-informed loss functions $\mathcal{L}_p$ is densely imposed and attains zero (De Ryck & Mishra, 2022; Mishra & Molinaro, 2023), we learn the solution of the PDE problem of arbitrary dimensions.

## 5 EXPERIMENTS

In this section, we present simulation results on estimating the recovery probability of a dynamical system which gives irregular ICBCs, and on estimating the solution of 3D Heat equations.

### 5.1 RECOVERY PROBABILITIES

We consider an autonomous system with dynamics

$$dx_t = u\, dt + dw_t, \tag{23}$$

where $x \in \mathbb{R}$ is the state, $w_t \in \mathbb{R}$ is the standard Wiener process with $w_0 = 0$, and $u \in \mathbb{R}$ is the system parameter. Given a set

$$\mathcal{C}_\alpha = \{x \in \mathbb{R} : x \geq \alpha\}, \tag{24}$$

we want to estimate the probability of reaching $\mathcal{C}_\alpha$ at least once within time horizon $t$ starting at some $x_0$. Here, $\alpha$ is the varying parameter of the set $\mathcal{C}_\alpha$. Mathematically this can be written as

$$s(x_0, t) := \mathbb{P}\left(\exists \tau \in [0,t], \text{ s.t. } x_\tau \in \mathcal{C}_\alpha \mid x_0\right). \tag{25}$$

From (Chern et al., 2021) we know that such probability is the solution of convection-diffusion equations with certain initial and boundary conditions

$$\textbf{PDE:} \quad \frac{\partial s}{\partial t}(x,t) - u\frac{\partial s}{\partial x}(x,t) - \frac{1}{2}\text{tr}\left(\frac{\partial^2 s}{\partial x^2}(x,t)\right) = 0, \ \forall [x,t] \in \mathcal{C}_\alpha^c \times \mathcal{T} \tag{26}$$

$$\textbf{ICBC:} \quad s(\alpha, t) = 1, \forall t \in \mathcal{T}, \quad s(x, 0) = 0, \forall x \in \mathcal{C}_\alpha^c, \tag{27}$$

where $\mathcal{C}_\alpha^c$ is the complement of $\mathcal{C}_\alpha$, and $\mathcal{T} = [0, T]$ for some $T$ of interest. Note that the initial condition and boundary condition at $(x, t) = (\alpha, 0)$ is not continuous,[2] which imposes difficulty for learning the solutions.

We train PI-DBSN with 3-layer fully connected neural networks with ReLU activation on varying parameters $u \in [0, 2]$ and $\alpha \in [0, 4]$, and test on randomly selected parameters in the same domain. We compare PI-DBSN with physics-informed neural network (PINN) (Cuomo et al., 2022) and physics-informed DeepONet (PI-DeepONet) (Goswami et al., 2023) with similar NN configurations. Fig. 2 visualizes the prediction results. It can be seen that both PI-DBSN and PINN can approximate the ground truth value accurately, while PI-DeepONet fails to do so. The possible reason is that PI-

| Method | Computation Time (s) |
|---|---|
| PI-DBSN | **370.48** |
| PINN | 809.86 |
| PI-DeepONet | 1455.16 |

Table 1: Computation time in seconds.

DeepONet can hardly capture the initial and boundary conditions correctly when the parameter set is relatively large. Besides, with the vanilla implementation of PI-DeepONet, the training tends to be unstable, and special training schemes such as the ones mentioned in Lee & Shin (2024) might be needed for finer results. The mean squared error (MSE) of the prediction are $3.064 \cdot 10^{-4}$ (Proposed PI-DBSN), $4.323 \cdot 10^{-4}$ (PINN), and $1.807 \cdot 10^{-1}$ (PI-DeepONet).

---

[2]When on the boundary of the $\mathcal{C}_\alpha$, the recovery probability at horizon $t = 0$ is $s(\alpha, 0) = 1$, but close to the boundary with very small $t$ the recovery probability is $s(x, 0) = 0$.

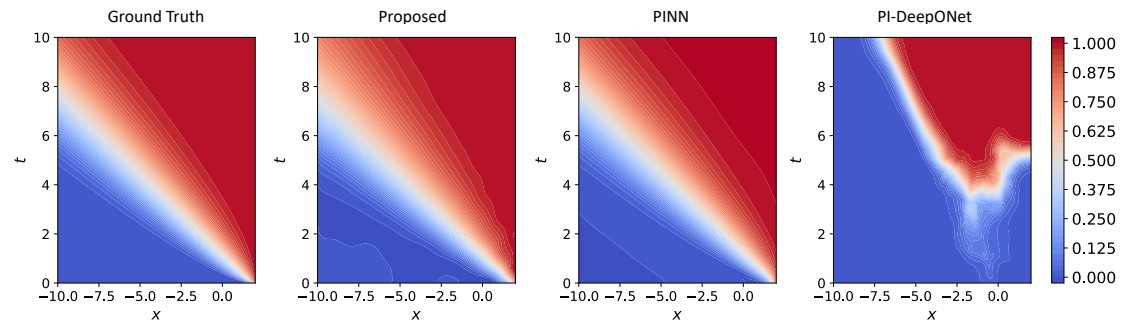

Figure 2: Recovery probability at $u = 1.5$ and $\alpha = 2$, $t \in [0, 10]$ is considered. The prediction MSE are $3.064 \cdot 10^{-4}$ (PI-DBSN), $4.323 \cdot 10^{-4}$ (PINN), and $1.807 \cdot 10^{-1}$ (PI-DeepONet).

| Number of Control Points | 2 | 5 | 10 | 15 | 20 | 25 |
|---|---|---|---|---|---|---|
| Number of NN Parameters | 4417 | 5392 | 9617 | 17092 | 27817 | 41792 |
| Training Time (s) | 241.76 | 223.53 | 247.39 | 295.67 | 310.83 | 370.48 |
| Prediction MSE ($\times 10^{-4}$) | 5357.9 | 7.327 | 7.313 | 5.817 | 4.490 | 3.064 |

Table 2: PI-DBSN prediction MSE with different numbers of control points along each dimension.

We then compare the training speed and computation time for the three methods, as shown in Fig. 3 and Table 1. We can see that the loss for PI-DBSN drops the fastest and reaches convergence in the shortest amount of time. This is because PI-DBSN has a relatively smaller NN size with the fixed B-spline basis, and achieves zero initial and boundary condition losses at the very beginning of the training. Besides, thanks to the analytical calculation of gradients and Hessians, the training time of PI-DBSN is the shortest among all three methods.

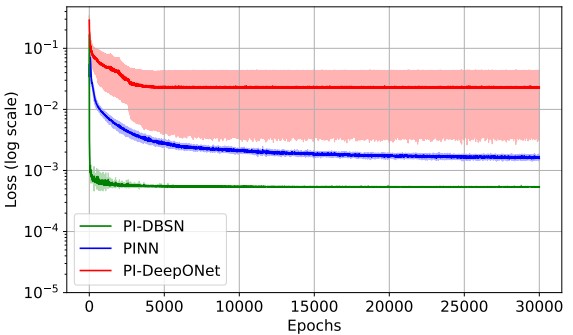

Figure 3: Total (physics and data) loss vs. epochs.

We also investigate the effect of the number of control points on the performance of PI-DBSN. Table 2 shows the approximation error and training time of PI-DBSN with different numbers of control points along each dimension. We can see that the training time increases as the number of control points increases, and the approximation error decreases, which matches with Theorem 4 which indicates more control points can result in less approximation error.

Experiment details and additional experiment results to verify the derivative calculations from B-splines and the optimality of the control points can be found in the Appendix of the paper.

## 5.2 3D HEAT EQUATIONS

We consider the 3D heat equation given by

$$\frac{\partial}{\partial t} s(x, t) = D \frac{\partial^2}{\partial x^2} s(x, t), \tag{28}$$

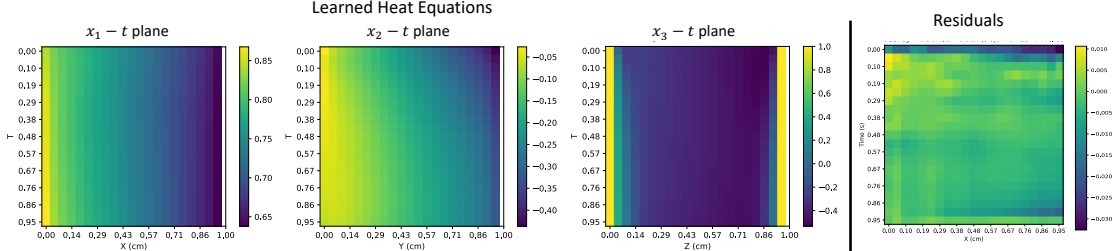

Figure 4: Evolution of 3D heat equation in a box with Dirichlet and Neumann boundary conditions. The learned solutions (left) and the residuals (right).

where $D = 0.1$ is the constant diffusion coefficient. Here $x = [x_1, x_2, x_3] \in \mathbb{R}^3$ are the states, and the domains of interest are $\Omega_{x_1} = \Omega_{x_2} = \Omega_{x_3} = [0, 1]$, and $\Omega_t = [0, 1]$. All lengths are in centimeters (cm) and the time is in seconds (s). In this experiment we solve equation 28 with random linear initial conditions:

$$s(x, t = 0) = \alpha_1 \cdot x_1 + \alpha_2 \cdot x_2 + \alpha_3 \cdot x_3 + \alpha_0 \tag{29}$$

where $\alpha_1, \alpha_2, \alpha_3 \in [-0.5, 0.5]$ and $\alpha_0 \in [0, 1]$ are randomly chosen. We impose the following Dirichlet and Neumann boundary conditions:

$$s(x, t | x_3 = 0) = s(x, t | x_3 = 1) = 1 \tag{30}$$

$$\frac{\partial}{\partial x_1} s(x, t | x_1 = 0) = \frac{\partial}{\partial x_1} s(x, t | x_1 = 1) = \frac{\partial}{\partial x_2} s(x, t | x_2 = 0) = \frac{\partial}{\partial x_2} s(x, t | x_2 = 1) = 0 \tag{31}$$

We train PI-DBSN on varying $\alpha$ with $\ell = 15$ control points along each dimension. Detailed training configurations can be found in the Appendix of the paper. Fig. 4 (left) shows the learned heat equation. It can be seen that the value is diffusing over time as intended. Fig. 4 (right) shows a slice of the residual of the learned heat equation in the $x_1$-$t$ plane. Although our initial condition does not adhere to the heat equation as estimated by the B-spline derivative, we quickly achieve a low residual. The average residuals during training and testing are 0.0028 and 0.0032, which indicates the efficacy of the PI-DBSN method.

## 6 CONCLUSION

In this paper, we consider the problem of learning solutions of PDEs with varying system parameters and initial and boundary conditions. We propose physics-informed deep B-spline networks (PI-DBSN), which incorporate B-spline functions into neural networks, to efficiently solve this problem. The advantages of the proposed PI-DBSN is that it can produce accurate analytical derivatives over automatic differentiation to calculate physics-informed losses, and can directly impose initial conditions and Dirichlet boundary conditions through B-spline coefficients. We prove theoretical guarantees that PI-DBSNs are universal approximators and under certain conditions can reconstruct PDEs of arbitrary dimensions. We then demonstrate in experiments that PI-DBSN performs better than existing methods on learning families of PDEs with discontinuous ICBCs, and has the capability of addressing higher dimensional problems.

For limitations and future work, we point out that even though B-splines are arguably a more efficient representation of the PDE problems, the PI-DBSN method still suffers from the curse of dimensionality. Specifically, the number of control points scales exponentially with the dimension of the problem, and as our theory and experiment suggest denser control points will help with obtaining lower approximation error. Besides, while the current formulation only allows regular geometry for the domain of interest, diffeomorphism transformations and non-uniform rational B-Splines (NURBS) (Piegl & Tiller, 2012) can be potentially applied to generalize the framework to irregular domains. How to further exploit the structure of the problem and learn large solution spaces in high dimensions with sparse data in complex domains are exciting future directions.

ACKNOWLEDGMENTS

Copyright 2024 Carnegie Mellon University and Duquesne University

This material is based upon work funded and supported by the Department of Defense under Contract No. FA8702-15-D-0002 with Carnegie Mellon University for the operation of the Software Engineering Institute, a federally funded research and development center.

NO WARRANTY. THIS CARNEGIE MELLON UNIVERSITY AND SOFTWARE ENGINEERING IN-STITUTE MATERIAL IS FURNISHED ON AN "AS-IS" BASIS. CARNEGIE MELLON UNIVER-SITY MAKES NO WARRANTIES OF ANY KIND, EITHER EXPRESSED OR IMPLIED, AS TO ANY MATTER INCLUDING, BUT NOT LIMITED TO, WARRANTY OF FITNESS FOR PURPOSE OR MERCHANTABILITY, EXCLUSIVITY, OR RESULTS OBTAINED FROM USE OF THE MATERIAL. CARNEGIE MELLON UNIVERSITY DOES NOT MAKE ANY WARRANTY OF ANY KIND WITH RESPECT TO FREEDOM FROM PATENT, TRADEMARK, OR COPYRIGHT INFRINGEMENT.

[DISTRIBUTION STATEMENT A] This material has been approved for public release and unlimited dis-tribution. Please see Copyright notice for non-US Government use and distribution.

This work is licensed under a Creative Commons Attribution-NonCommercial 4.0 International License. Requests for permission for non-licensed uses should be directed to the Software Engineering Institute at permission@sei.cmu.edu.

DM25-0126

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

# APPENDIX

# A    PROOF OF THEOREMS

## A.1    PROOF OF THEOREM 1

*Proof.* (Theorem 1) From (Jia & Lei, 1993; Strang & Fix, 1971) we know that given $d$ the least square spline approximation of $\hat{s}(x) = \sum_{i=1}^{\ell} c_i B_{i,d}(x)$ can be obtained by applying pre-filtering, sampling and post-filtering on $s$, with $L_2$ error bounded by

$$\|s - \hat{s}\|_2 \leq C_d \cdot T^d \cdot \|s^{(d)}\|, \tag{32}$$

where $C_d$ is a known constant (Blu & Unser, 1999), $T$ is the sampling interval of the pre-filtered function, and $\|s^{(d)}\|$ is the norm of the $d$-th derivative of $s$ defined by

$$\left\|s^{(d)}\right\| = \left(\frac{1}{2\pi} \int_{-\infty}^{+\infty} \omega^{2d} |S(\omega)|^2 d\omega\right)^{1/2}, \tag{33}$$

and $S(\omega)$ is the Fourier transform of $s(x)$. Note that given $s$ and $d$, $\left\|s^{(d)}\right\|$ is a known constant.

Then, from (Unser, 1999) we know that the samples from the pre-filtered functions are exactly the control points $c_i$ that minimize the $L_2$ norm in equation 15 in our problem. In other words, the sampling time $T$ and the number of control points $\ell$ are coupled through the following relationship

$$T = \frac{b-a}{\ell-1}, \tag{34}$$

since the domain is $[a, b]$ and it is divided into $\ell - 1$ equispaced intervals for control points. Then with $c_i$ being the samples with interval $T$, we can rewrite the error bound into

$$\|s - \hat{s}\|_2 \leq C_d \cdot \left(\frac{b-a}{\ell-1}\right)^d \cdot \|s^{(d)}\| \tag{35}$$

Thus we know that for $\forall \epsilon > 0$, we can find $\ell$ such that

$$\|s - \hat{s}\|_2 \leq \frac{(b-a)^d C_d \|s^{(d)}\|}{(\ell-1)^d} \leq \epsilon \tag{36}$$

because for fixed $d$ the numerator is a constant, and the $L_2$ norm bound converges to 0 as $\ell \to \infty$. □

## A.2    PROOF OF LEMMA 2

*Proof.* (Lemma 2) For given $\ell_1, \cdots, \ell_n$, let $C := [c]_{\ell_1 \times \cdots \times \ell_n}$ be the control points tensor such that $\|s(x_1, x_2, \cdots, x_n) - \hat{s}(x_1, x_2, \cdots, x_n)\|_2$ is minimized. Let $(x'_1, x'_2, \cdots, x'_n)$ denote the knot points in

the $n$-dimensional space, *i.e.,* the equispaced grids where the control points are located. Then from Theorem 1 and the separability of the B-splines (Pratt, 2007), we know that

$$\int_{a_1}^{b_1} (s - \hat{s})(s - \hat{s})^*(x_1, x_2', \cdots, x_n')dx_1 \leq \epsilon_{x_1}, \tag{37}$$

where $\epsilon_{x_1} = \frac{(b-a)^{d_1}C_{d_1}\|s^{(d_1)}\|}{(\ell_1-1)^{d_1}}$. This shows that the $L_2$ norm along the $x_1$ direction at any knots points $(x_2', \cdots, x_n')$ is bounded. Now we show the following is bounded

$$\int_{a_2}^{b_2} \int_{a_1}^{b_1} (s - \hat{s})(s - \hat{s})^*(x_1, x_2, x_3', \cdots, x_n')dx_1 dx_2. \tag{38}$$

We argue that $s$ is Lipschitz as it is defined on a bounded domain and is $d$-time differentiable, and $\hat{s}$ is also Lipschitz as B-spline functions of any order are Lipschitz (Prautzsch, 2002; Kunoth et al., 2018) and $C$ is finite. Then we know that $(s - \hat{s})(s - \hat{s})^*$ is Lipschitz with some Lipschitz constant $L_{x_i}$ along dimension $i$ for $i = 1, 2, \cdots, n$. For $\forall x_2 \in [a_2, b_2]$, there is a knot point $x_2'$ such that $|x_2 - x_2'| \leq \frac{b_2-a_2}{\ell_2-1}$ since knot points are equispaced. Thus, we know for $\forall x_2 \in [a_2, b_2]$, there is $x_2'$ such that

$$|(s - \hat{s})(s - \hat{s})^*(x_1, x_2, x_3', \cdots, x_n') - (s - \hat{s})(s - \hat{s})^*(x_1, x_2', x_3', \cdots, x_n')| \leq L_{x_2}\frac{b_2 - a_2}{\ell_2 - 1} \tag{39}$$

Then we have

$$\int_{a_2}^{b_2} \int_{a_1}^{b_1} (s - \hat{s})(s - \hat{s})^*(x_1, x_2, x_3', \cdots, x_n')dx_1 dx_2 \tag{40}$$

$$\leq \int_{a_2}^{b_2} \int_{a_1}^{b_1} (s - \hat{s})(s - \hat{s})^*(x_1, x_2', x_3', \cdots, x_n')dx_1 dx_2$$

$$+ \int_{a_2}^{b_2} \int_{a_1}^{b_1} |(s - \hat{s})(s - \hat{s})^*(x_1, x_2, x_3', \cdots, x_n') - (s - \hat{s})(s - \hat{s})^*(x_1, x_2', x_3', \cdots, x_n')|dx_1 dx_2$$

$$\tag{41}$$

$$\leq \int_{a_2}^{b_2} \int_{a_1}^{b_1} (s - \hat{s})(s - \hat{s})^*(x_1, x_2', x_3', \cdots, x_n')dx_1 dx_2 + \int_{a_2}^{b_2} \int_{a_1}^{b_1} L_{x_2}\frac{b_2 - a_2}{\ell_2 - 1}dx_1 dx_2 \tag{42}$$

$$\leq (b_2 - a_2)\left[\epsilon_{x_1} + L_{x_2}\frac{(b_2 - a_2)(b_1 - a_1)}{\ell_2 - 1}\right] := \epsilon_{x_1, x_2}, \tag{43}$$

where equation 41 is the triangle inequality of norms, and equation 42 is due to the Lipschitz-ness of the function.

Similarly we can show the bound when we integrate the next dimension

$$\int_{a_3}^{b_3} \int_{a_2}^{b_2} \int_{a_1}^{b_1} (s - \hat{s})(s - \hat{s})^*(x_1, x_2, x_3, x_4', \cdots, x_n') dx_1 dx_2 dx_3 \tag{44}$$

$$\leq \int_{a_3}^{b_3} \int_{a_2}^{b_2} \int_{a_1}^{b_1} (s - \hat{s})(s - \hat{s})^*(x_1, x_2, x_3', x_4', \cdots, x_n') dx_1 dx_2 dx_3$$

$$+ \int_{a_3}^{b_3} \int_{a_2}^{b_2} \int_{a_1}^{b_1} |(s - \hat{s})(s - \hat{s})^*(x_1, x_2, x_3, x_4', \cdots, x_n') - (s - \hat{s})(s - \hat{s})^*(x_1, x_2, x_3', x_4', \cdots, x_n')| dx_1 dx_2 dx_3 \tag{45}$$

$$\leq \int_{a_3}^{b_3} \int_{a_2}^{b_2} \int_{a_1}^{b_1} (s - \hat{s})(s - \hat{s})^*(x_1, x_2, x_3', x_4', \cdots, x_n') dx_1 dx_2 dx_3 + \int_{a_3}^{b_3} \int_{a_2}^{b_2} \int_{a_1}^{b_1} L_{x_3} \frac{b_3 - a_3}{\ell_3 - 1} dx_1 dx_2 dx_3 \tag{46}$$

$$\leq (b_3 - a_3) \left[ \epsilon_{x_1, x_2} + L_{x_3} \frac{(b_3 - a_3)(b_2 - a_2)(b_1 - a_1)}{\ell_3 - 1} \right] := \epsilon_{x_1, x_2, x_3}. \tag{47}$$

We know that $\epsilon_{x_1, x_2, x_3} \to 0$ when $\ell_i \to \infty$ for $i = 1, 2, 3$. By keeping doing this, recursively we can find the bound $\epsilon_{x_1, \cdots, x_n}$ that

$$\int_{a_n}^{b_n} \cdots \int_{a_1}^{b_1} (s - \hat{s})(s - \hat{s})^*(x_1, \cdots, x_n) dx_1 \cdots dx_n \leq \epsilon_{x_1, \cdots, x_n}, \tag{48}$$

where the left hand side is exactly $\|s(x_1, x_2, \cdots, x_n) - \hat{s}(x_1, x_2, \cdots, x_n)\|_2^2$, and the right hand side $\epsilon_{x_1, \cdots, x_n} \to 0$ when $\ell_i \to \infty$ for all $i = 1, 2, \cdots, n$. Thus for any $\epsilon > 0$, we can find $\ell_i$ for $i = 1, 2, \cdots, n$ such that

$$\|s(x_1, x_2, \cdots, x_n) - \hat{s}(x_1, x_2, \cdots, x_n)\|_2 \leq \epsilon \tag{49}$$

$\square$

### A.3 PROOF OF THEOREM 4

*Proof.* (Theorem 4) For any $u$ and $\alpha$, from Lemma 2 we know that there is $\ell_1, \cdots, \ell_n$ and the control points realization $C := [c]_{\ell_1 \times \cdots \times \ell_n}$ such that $\|s(x_1, x_2, \cdots, x_n) - \hat{s}(x_1, x_2, \cdots, x_n)\|_2 \leq \epsilon_1$ for any $\epsilon_1 > 0$, where $\hat{s}$ is the B-spline approximation defined in equation 11 with the control points tensor $C$. Then, from Theorem 3 we know that there is a DBSN configuration $G_{\boldsymbol{\theta}}(u, \alpha)$ and corresponding parameters $\boldsymbol{\theta}^*$ such that $\|G_{\boldsymbol{\theta}^*}(u, \alpha) - C\| \leq \epsilon_2$ for any $\epsilon_2 > 0$. Since B-spline functions of any order are continuous and Lipschitz (Prautzsch, 2002; Kunoth et al., 2018), we know that $\|\tilde{s} - \hat{s}\|_2 \leq L\epsilon_2$ for some Lipschitz related constant $L$. Then by triangle inequality of the $L_2$ norm, we have

$$\|\tilde{s} - s\|_2 \leq \|\tilde{s} - \hat{s}\|_2 + \|\hat{s} - s\|_2 \leq \epsilon_1 + L\epsilon_2. \tag{50}$$

For any $\epsilon > 0$ we can find $\epsilon_1$ and $\epsilon_2$ such that $\epsilon = \epsilon_1 + L\epsilon_2$ to bound the norm. $\square$

## B ADDITIONAL THEORETICAL RESULTS

Considering a one-dimensional B-spline of the form as equation 8, where $x \in [a, b]$, we have

$$\hat{s} \in [a, b] \times [\underline{c}, \overline{c}], \tag{51}$$

where

$$\underline{c} = \min_{i=1,\ldots,\ell} c_i, \qquad \overline{c} = \max_{i=1,\ldots,\ell} c_i.$$

This property is inherent to the Bernstein polynomials used to generate Bézier curves. Specifically, the Bézier curve is a subtype of the B-spline, and it is also possible to transform Bézier curves into B-splines and vice versa (Prautzsch, 2002).

This property also holds in the multidimensional case when the B-spline is represented by a tensor product of the B-spline basis functions in equation 11 (Prautzsch, 2002):

$$\hat{s} \in [a_1, b_1] \times \cdots \times [a_n, b_n] \times [\underline{c}, \overline{c}], \tag{52}$$

where

$$\underline{c} = \min_{\substack{i_1=1,\ldots,\ell_1 \\ i_2=1,\ldots,\ell_2 \\ \vdots \\ i_n=1,\ldots,\ell_n}} c_{i_1,i_2,\ldots,i_n}, \qquad \overline{c} = \max_{\substack{i_1=1,\ldots,\ell_1 \\ i_2=1,\ldots,\ell_2 \\ \vdots \\ i_n=1,\ldots,\ell_n}} c_{i_1,i_2,\ldots,i_n}.$$

This property offers a practical tool for verifying the reliability of the results produced by the trained learning scheme. In the case of learning recovery probabilities, the approximated solution should provide values between $0$ and $1$. Since the number of control points is finite, a robust and reliable solution occurs if all generated control points are within the range $[0, 1]$, *i.e.,*

$$\underline{c} = 0 \qquad \overline{c} = 1.$$

## C  EXPERIMENT DETAILS

### C.1  TRAINING DATA

**Recovery Probabilities:**  The convection diffusion PDE defined in equation 26 and equation 27 has analytical solution

$$s(x,t) = \int_0^t \frac{(\alpha - x)}{\sqrt{2\pi\tau^3}} \exp\left(-\frac{((\alpha - x) - u\tau)^2}{2\tau}\right) d\tau, \tag{53}$$

where $\alpha$ is the parameter of the boundary of the set in equation 24, and $u$ is the parameter of the system dynamics in equation 23. We use numerical integration to solve equation 53 to obtain ground truth training data for the experiments.

### C.2  NETWORK CONFIGURATIONS

**Recovery Probabilities:**  For PI-DBSN and PINN, we use 3-layer fully connected neural networks with ReLU activation functions. The number of neurons for each hidden layer is set to be $64$. For PI-DeepONet, we use 3-layer fully connected neural networks with ReLU activation functions for both the branch net and the trunk net. The number of neurons for each hidden layer is set to be $64$. All methods use Adam as the optimizer.

**3D Heat Equations:**  We set the B-splines to have the same number $\ell = 15$ of equispaced control points in each direction including time. We sample the solution of the heat equation at 21 equally spaced locations in each dimension. Thus, each time step consists of $15^3 = 3375$ control points and each sample returns $15^4 = 50625$ control points total. The inputs to our neural network are the values of $\alpha$ from which it learns the control points, and subsequently the initial condition surface via direct supervised learning. This is followed by learning the control points associated with later times, ($t > 0$) via the PI-DBSN method. Because of the natural time evolution component of this problem, we use a network with residual connections and sequentially learn each time step. The neural network has a size of about $5 \times 10^4$ learnable parameters.

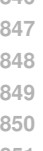

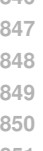

Figure 5: Physics loss vs. epochs.

### C.3 TRAINING CONFIGURATIONS

All comparison experiments are run on a Linux machine with Intel i7 CPU and 16GB memory.

### C.4 EVALUATION METRICS

The reported mean square error (MSE) is calculated on the mesh grid of the domain of interest. Specifically, for the recovery probability experiment, the testing data is generated and the prediction is evaluated on $(x, t) \in [-10, \alpha] \times [0, 10]$ with $dx = 0.1$ and $dt = 0.1$. For the 3D heat equation problem, the testing evaluation is on $(x_1, x_2, x_3, t) \in [0, 1]^4$ with $dx = dt = 0.01$.

The $|\cdot|$ used in evaluating data and physics losses denote absolute values.

### C.5 LOSS FUNCTION VALUES

We visualize the physics loss and data loss separately for all three methods considered in section 5.1. Fig. 5 shows the physics loss and Fig. 6 shows the data loss (without ICBC losses for fair comparison with PI-DBSN). We can see that PI-DBSN achieves similar physics loss values compared with PINN, but converges much faster. Besides, PI-DBSN achieves much lower data losses under this varying parameter setting, possibly due to its efficient representation of the solution space. PI-DeepONet has high physics and data loss values in this case study.

### C.6 PINN PERFORMANCE ON 3D HEAT EQUATIONS

We report results of PINN (Raissi et al., 2019) for the 3D heat equations case study in section 5.2 for comparison. The PINN consists of 4 hidden layers with 50 neurons in each layer. We use Tanh as the activation functions. We train PINN for 30000 epochs, with physics and data loss weights $w_p = w_d = 1$. Fig. 7 visualizes the PINN prediction along different planes. The testing residual is 0.0121, which is higher than the reported value (0.0032) for PI-DBSN.

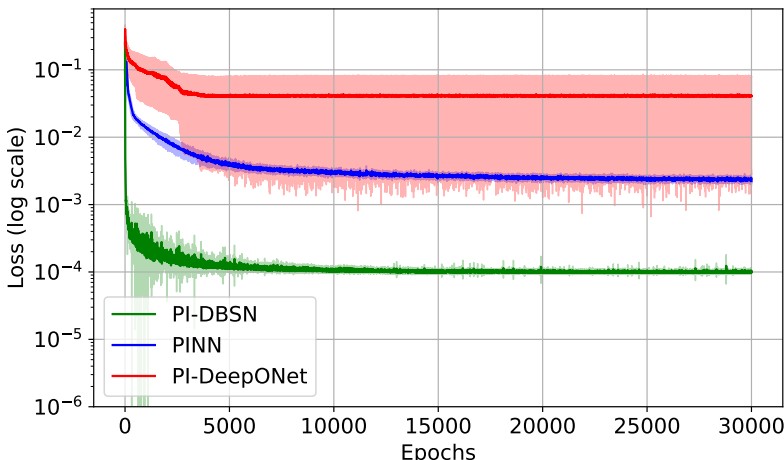

Figure 6: Data loss vs. epochs.

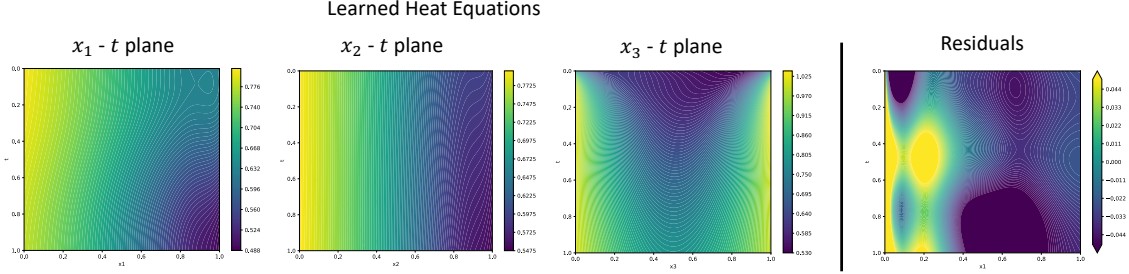

Figure 7: The learned solutions (left) and the residuals (right) for the 3D heat equations with PINN.

## D    ADDITIONAL EXPERIMENTS

### D.1    B-SPLINE DERIVATIVES

In this section, we show that the analytical formula in equation 12 can produce fast and accurate calculation of B-spline derivatives. Fig. 8 shows the derivatives from B-spline analytical formula and finite difference for the 2D space $[-10, 2] \times [0, 10]$ with the number of control point $\ell_1 = \ell_2 = 15$. The control points are generated randomly on the 2D space, and the derivatives are evaluated at mesh grids with $N_1 = N_2 = 100$. We can see that the derivatives generated from B-spline formulas match well with the ones from finite difference, except for the boundary where finite difference is not accurate due to the lack of neighboring data points.

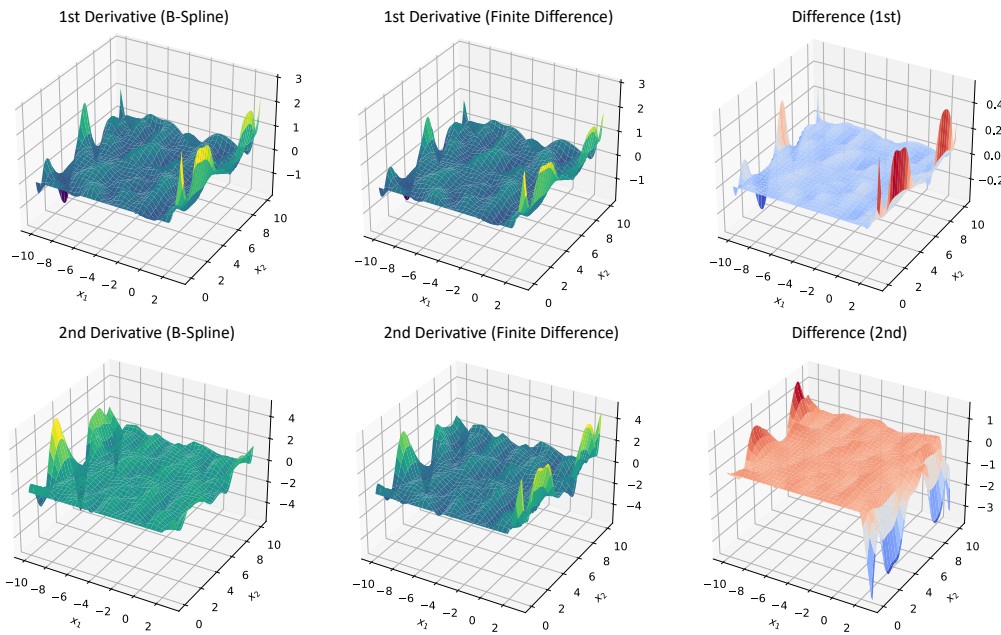

Figure 8: First and second derivatives from B-splines and finite difference.

### D.2    OPTIMALITY OF CONTROL POINTS

In this section, we show that the learned control points of PI-DBSN are near-optimal in the $L_2$ norm sense. For the recovery probability problem considered in section 5.1, we investigate the case for a fixed set of system and ICBC parameters $u = 1.5$ and $\alpha = 2$. We use the number of control points $\ell_1 = \ell_2 = 25$ on the domain $[-10, 2] \times [0, 10]$, and obtain the optimal control points $C^*$ in the $L_2$ norm sense by solving least square problem (Deng & Lin, 2014) with the ground truth data. We then compare the learned control points $C$ with $C^*$ and the results are visualized in Fig. 9. We can see that the learned control points are very close to the optimal control points, which validates the efficacy of PI-DBSN. The only region where the difference is relatively large is near $c_{25,0}$, where the solution is not continuous and hard to characterize with this number of control points.

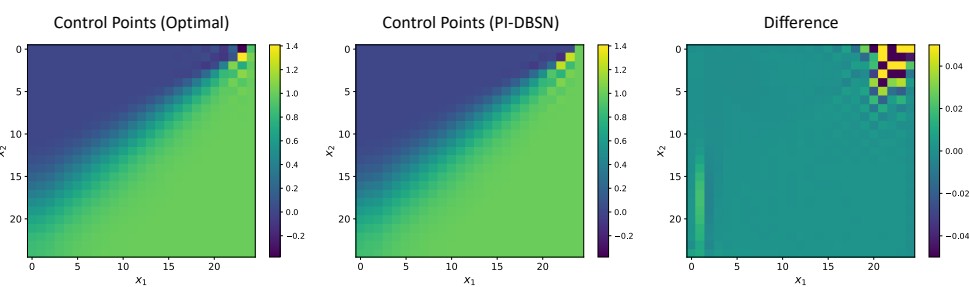

Figure 9: Control points.

| Method | Computation Time (s) |
|--------|---------------------|
| PI-DBSN | **271** |
| PINN | 365 |
| PI-DeepONet | 429 |

Table 3: Computation time in seconds (with A100 GPU).

## D.3 EXPERIMENTS ON GPUs

We tested the performance of PI-DBSN and the baselines on a cloud server with one A100 GPU. Note that our implementations are in PyTorch (Paszke et al., 2019), thus it naturally adapts to both CPU and GPU running configurations. The experiment settings are the same as in section 5.1. The running time of the three methods are reported in Table 3. We can see that GPU implementation accelerates training for all three methods, and PI-DBSN has the shortest running time, which is consistent with the CPU implementation results.

## D.4 ROBUSTNESS AND LOSS FUNCTION WEIGHTS ABLATIONS

In this section, we provide ablation experiments of the proposed PI-DBSN with different loss function configurations, and examine its robustness again noise. The setting is described in section 5.1. We first train with noiseless data and vary the data loss weight $w_d$. Table 4 shows the average MSE and its standard deviation over 10 independent runs. We can see that with more weights on the data loss, the prediction MSE reduces as noiseless data help with PI-DBSN to learn the ground truth solution. We then train with injected additive zero-mean Gaussian noise with standard deviation 0.05 and vary the physics loss weight $w_p$. Table 5 shows the results. It can be seen that increasing physics loss weights help PI-DBSN to learn the correct neighboring relationships despite noisy training data, which reduces prediction MSE. In general, the weight choices should depend on the quality of the data, the training configurations (*e.g.,* learning rates, optimizer, neural network architecture).

| $w_d$ | 1 | 2 | 3 | 4 | 5 |
|-------|---|---|---|---|---|
| $w_p$ | 1 | 1 | 1 | 1 | 1 |
| Prediction MSE ($\times 10^{-5}$) | $36.76 \pm 12.16$ | $12.91 \pm 10.40$ | $10.21 \pm 3.99$ | $9.28 \pm 6.78$ | $3.95 \pm 1.36$ |

Table 4: PI-DBSN prediction MSE (noiseless data).

| $w_d$ | 1 | 1 | 1 | 1 | 1 |
|---|---|---|---|---|---|
| $w_p$ | 1 | 2 | 3 | 4 | 5 |
| Prediction MSE ($\times 10^{-4}$) | $31.58 \pm 6.46$ | $33.15 \pm 7.77$ | $13.37 \pm 11.74$ | $7.95 \pm 6.24$ | $3.86 \pm 2.05$ |

Table 5: PI-DBSN prediction MSE (additive Gaussian noise data).

| Number of Hidden Layers | 2 | 3 | 4 | 5 |
|---|---|---|---|---|
| Number of NN parameters | 37632 | 41792 | 45952 | 50112 |
| Prediction MSE ($\times 10^{-4}$) | $1.12 \pm 0.43$ | $0.90 \pm 0.42$ | $3.17 \pm 2.46$ | $3.12 \pm 2.81$ |

Table 6: PI-DBSN prediction MSE with different numbers of NN layers.

### D.5 NUMBER OF NN LAYERS AND PARAMETERS ABLATION

In this section, we show ablation results on the number of neural network (NN) layers and parameters. We follow the experiment settings in section 5.1, and train the proposed PI-DBSN with different numbers of hidden layers, each with 10 independent runs. The number of NN parameters, the prediction MSE and its standard deviation are shown in Table 6. We can see that with 3 layers the network achieves the lowest prediction errors, while the number of layers does not have huge influence on the overall performance.

### D.6 BURGERS' EQUATION

We conduct additional experiments on the following Burgers' equation.

$$\frac{\partial s}{\partial t} + us\frac{\partial s}{\partial x} = \nu\frac{\partial^2 s}{\partial x^2}, \tag{54}$$

where $\nu = 0.01$ and $u \in [0.5, 1.5]$ is a changing parameter. The domain of interest is set to be $(x, t) \in [0, 10] \times [0, 8]$, and the initial condition is

$$s(x, 0) = \exp\{-(x - \alpha)^2/2\}, \tag{55}$$

where $\alpha \in [2, 4]$ is a changing parameter. We train PI-DBSN with 3-layer fully connected neural networks with ReLU activation on varying parameters $u \in [0.5, 1.5]$ and $\alpha \in [2, 4]$, and test on randomly selected parameters in the same domain. The B-spline basis of order 4 is used and the number of control points along $x$ and $t$ are set to be $\ell_x = \ell_t = 100$. Note that more control points are used in this case study compared to the convection diffusion equation in section 5.1, as the solution of the Burgers' equation has higher frequency along the ridge which requires finer control points to represent. Fig. 10 visualizes the prediction results on several random parameter settings. The average MSE across 20 test cases is $1.319 \pm 0.408 \times 10^{-2}$. This error rate is comparable to the Fourier neural operators as reported in (Li et al., 2020, Figure 3).

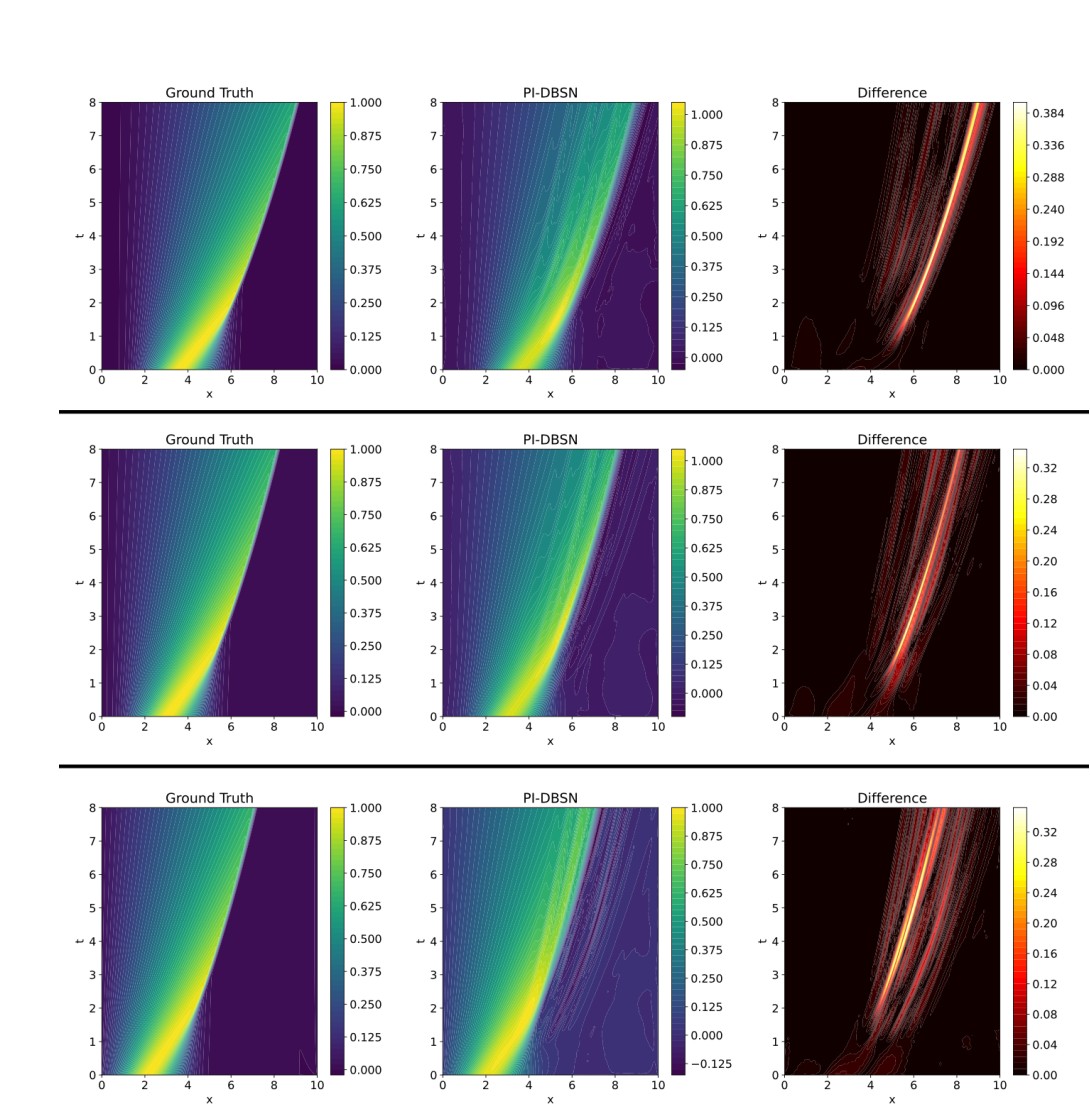

Figure 10: Results on Burgers' equations with different random parameter settings.

