# OpenReview forum: "Physics-Informed Deep B-Spline Networks"
_ICLR.cc/2025/Conference — Submitted to ICLR 2025_

### Official Review · Reviewer_KSqP · 2024-10-28

**Soundness:** 4
**Presentation:** 4
**Contribution:** 3
**Rating:** 8
**Confidence:** 3

**Summary:**

This paper proposes a physics framework for learning B-spline control points in systems with varying initial and boundary conditions. The approach allows for using analytical expressions as physics informed losses since B-spline function gradients can be expressed as analytical expressions. The proposed approach is used for estimating high dimensional surfaces described by families of ODEs/PDEs. The authors provide theoretical bounds and empirical evidence for their claims.

**Strengths:**

Originality: This paper has good mathematical intuition and a novel way to estimate the solution of PDEs using B-splines. This approach makes it easier to use PINN loses using analytical expressions improving efficiency over autograd.  The use of control points allows for any Dirichlet boundary conditions and initial conditions to be directly assigned.

Quality: The method is very clearly described, and the authors provide a good mathematical background. They clearly state the difference between their proposed approach and KAN based approaches. The paper provides theoretical bounds and empirical evidence.

Significance: This work has a lot of potential in the field of scientific ML, as the authors show significant speedup over Physics informed ML techniques, and a reasonable improvement in accuracy of predictions.

**Weaknesses:**

While the authors provide a good theoretical basis for B-spline based PINNs, the experimental section is quite small. They evaluate on 2 settings. However, it would be good to showcase the performance on other standard evaluation benchmarks (Navier-Stokes, Darcy Flow, Burgers, Kolmogorov flow etc.)

The use of splines to train PINNs is not new. [1] uses Hermite splines to approximate PDEs. [2], [3] proposes using neural networks to learn the coefficients of B-spline functions. It would be good to show the differences between the proposed approach and these works.

[1] Wandel, Nils, et al. "Spline-pinn: Approaching pdes without data using fast, physics-informed hermite-spline cnns." Proceedings of the AAAI conference on artificial intelligence. Vol. 36. No. 8. 2022.

[2] Doległo, Kamil, et al. "Deep neural networks for smooth approximation of physics with higher order and continuity B-spline base functions." arXiv preprint arXiv:2201.00904 (2022).

[3] Zhu, Xuedong, et al. "A Best-Fitting B-Spline Neural Network Approach to the Prediction of Advection–Diffusion Physical Fields with Absorption and Source Terms." Entropy 26.7 (2024): 577.

**Questions:**

1.	How does the proposed approach compare against [1], [2], [3]?

[1] Wandel, Nils, et al. "Spline-pinn: Approaching pdes without data using fast, physics-informed hermite-spline cnns." Proceedings of the AAAI conference on artificial intelligence. Vol. 36. No. 8. 2022.

[2] Doległo, Kamil, et al. "Deep neural networks for smooth approximation of physics with higher order and continuity B-spline base functions." arXiv preprint arXiv:2201.00904 (2022).

[3] Zhu, Xuedong, et al. "A Best-Fitting B-Spline Neural Network Approach to the Prediction of Advection–Diffusion Physical Fields with Absorption and Source Terms." Entropy 26.7 (2024): 577.

---

> ### Author Response · Authors · 2024-11-21
>
> Thank you for your comments.
>
> Q1: While the authors provide a good theoretical basis for B-spline based PINNs, the experimental section is quite small. They evaluate on 2 settings. However, it would be good to showcase the performance on other standard evaluation benchmarks (Navier-Stokes, Darcy Flow, Burgers, Kolmogorov flow etc.)
>
> A1: We have added additional experiment results on Burgers' equation in Appendix D.6. The results of PI-DBSN are comparable to Fourier Neural Operators as reported in [4], in terms of prediction error.
>
> Q2: How does the proposed approach compare against [1], [2], [3]?
>
> [1] Wandel, Nils, et al. "Spline-pinn: Approaching pdes without data using fast, physics-informed hermite-spline cnns." Proceedings of the AAAI conference on artificial intelligence. Vol. 36. No. 8. 2022.
>
> [2] Doległo, Kamil, et al. "Deep neural networks for smooth approximation of physics with higher order and continuity B-spline base functions." arXiv preprint arXiv:2201.00904 (2022).
>
> [3] Zhu, Xuedong, et al. "A Best-Fitting B-Spline Neural Network Approach to the Prediction of Advection–Diffusion Physical Fields with Absorption and Source Terms." Entropy 26.7 (2024): 577.
>
> A2: The proposed approach is different from the mentioned existing literature in the following ways.
>
> For [1], firstly, Hermite splines and B-splines are two different methods for different problems. In general, Hermite splines are for interpolations, while B-splines are for approximations. The proposed method is different from [1] as we leverage B-spline properties to address an approximation problem. Secondly, with Hermite splines, one needs to “choose the right spline order $n$ to be able to compute the physics-informed loss for a given PDE” according to [1], while we do not have such issues. Finally, [1] proposes to use CNN to learn mappings from control points at time $t$ to control points at $t+1$, while we directly learn control points for the entire state-time domain.
>
> For [2] and [3], both papers learn B-splines to approximate PDEs, but they only consider data fitting problems, where no physics model loss is imposed in the learning process. Instead, we propose physics-informed deep B-spline networks and show that the analytical calculation of derivatives for physics model losses and direct assignment of ICBCs are advantageous over existing physic-informed learning methods.
>
>
> [4] Li, Zongyi, et al. "Fourier neural operator for parametric partial differential equations." _arXiv preprint arXiv:2010.08895_ (2020).

---

> > ### Comment · Reviewer_KSqP · 2024-11-22
> >
> > Thank you for providing the clarification and the results on Burgers equation. I will retain the score.

---

### Official Review · Reviewer_dWHg · 2024-11-01

**Soundness:** 2
**Presentation:** 3
**Contribution:** 2
**Rating:** 6
**Confidence:** 4

**Summary:**

This paper proposes PI-DBSN, a direct neural network solution to learn to solve parametric PDEs with different initial or boundary conditions. PI-DBSN directly learns the weight of B-spline basis functions, by training under both data loss and physics-informed loss. Authors address that there are two-fold advantages of learning spline interpolants. On the one hand, universal approximation theorem holds for any solution in $L^2$ domain. On the other hand, it can solve problems under multiple configurations of initial or boundary conditions and the geometry of the domain has the potential to be dynamically change based on its parameterization.

**Strengths:**

This paper has an unique viewpoint as it trains a neural network which maps both physical parameters in PDE setup and parametric domain into weights of B-spline basis functions. Manipulating an NN solver to adapt to geometry of domain is a challenging topic. Moreover, an extensive discussion of related works supports its novelty and differentiate this paper from many previous researches. Authors provide a through analysis over universal approximation theorem as well.

**Weaknesses:**

While I am expected to see some further discussions of how to learn a PDE solution under different configurations of IC/BCs by fitting spline interpolants, the design of experiments hinder a reader to appreciate PI-DBSN.  Authors show performance for a convection-diffusion equation in space-time domain and a 3D heat equation, but on a regular box-domain. Even for this simple setup I didn't see superiority of applying PI-DBSN, rather than using PINN, let alone several NO approaches was not compared. In second case where a 3D heat equation was learnt, author even did not compare with any other methods, and $10^{-3}$ scale of residual error is not a wow number.

Though not required due to extensive work of experiments, I wish to, at least hear from authors, if complex geometric domain, such as $L$-shape problem introduced in Doleglo et al. (2022), or hyper-elastic problems setup in [Geo-FNO](https://arxiv.org/pdf/2207.05209) can be computed via PI-DBSN. Under these setups, is there any potential challenges or modifications needed to apply PI-DBSN to irregular geometries, and how the spline basis representation might need to be adapted.

Besides simple setup in experiments, several statements in this paper is inappropriate. Please consider revise them. For instance:


*    line 080: The term "Dirichlet initial and boundary conditions" is incorrect. It should be "Dirichlet boundary conditions and initial conditions".
*    line 112-113: It claims Neural Operator solution "do not account for varying initial and boundary conditions". Nevertheless, in original FNO paper, even for Burger's euqation, initial conditions are drawn from random samples.
*    line 246-247: "initial condition is given by $s(0,t)=0, \forall t$" is an obvious typo. It should be  $s(x,0)=0, \forall x$. Even corrected, I don't see why a constant boundary shows superiority of spline basis representations. A correct and complete statement shall point out that: even under this simple boundary setup, prior works like PINN would fail to comply their solutions with respect to this particular BC.
*    Line 338: "3D Fokker-Planck Equation" is inconsistent with section title "Heat Dispersion in a 3D box" (line 418), and setup "3D heat equation" (line 420), even though they are related.
*    All norms in loss functions $\mathcal{L}$ are of form $\lvert \cdot \rvert$ (line 223-229). Is this $l^2$ norm or $L^2$ norm or absolute value? Please specify.

**Questions:**

Some other questions, if answered, can improve the quality of this paper:

*    Figure 3 indicates PINN attains average better performance. It is also indicated in line 387-388 that prediction MSE of PI-DBSN is worse compared with PINN. Can you provide more details of network setup, and explain "similar NN configuration" stated in line 364-365. How many parameters of NN in each setup and how computation time would change if one set up same or similar number of parameters in each experiment.
*    What is the residual error in convection-diffusion setups?
*    How does a plot of number of NN parameters v.s. MSE. looks like? In table  2 only the relationship between number of control points v.s. MSE is reported.
*    Are all experiments conducted over noiseless setup? Is PI-DBSN robust against noisy observation?
*    What is the performance of PI-DBSN over Burger's equation and Navier-Stokes equation using the same setup in FNO paper? Are there any specific reasons that the comparison ruled out FNO branch of method?
*    How is physics model loss being evaluated. Namely, what is the quadrature rule when computing $\lvert\mathcal{F}_i(s,x,u)\rvert^2$?
*    Is there any ablation study on how to choose $\omega_p$ and $\omega_d$ to attain better performance? In line 230, authors claim both are set to $1$.

---

> ### Author Response · Authors · 2024-11-21
>
> Thank you for your comments. We have revised the paper to correct the typos and ambiguous claims you raised.
>
> Q1: Figure 3 indicates PINN attains average better performance. It is also indicated in line 387-388 that prediction MSE of PI-DBSN is worse compared with PINN. Can you provide more details of network setup, and explain "similar NN configuration" stated in line 364-365. How many parameters of NN in each setup and how computation time would change if one set up same or similar number of parameters in each experiment.
>
> A1: The loss functions of PINN and PI-DBSN are slightly different, as PINN takes initial condition and boundary condition (ICBC) losses as well, while PI-DBSN can directly specify ICBC values. The loss plot in Figure 3 is normalized, thus PINN appears to have lower overall loss values. From the prediction results we know that the test performance of PINN and PI-DBSN are comparable, and note that PI-DBSN (MSE $3.064 \cdot 10^{-4}$) has slightly better performance than PINN (MSE $4.323 \cdot 10^{-4}$). We use exactly the same NN architecture for PI-DBSN and PINN (3-layer fully connected NN with ReLU activation functions, with $64$ neurons in each hidden layer), and the same NN for both trunk net and branch net for PI-DeepONet. We have also added NN size ablation experiments in Appendix D.5.
>
> Q2: What is the residual error in convection-diffusion setups? All norms in loss functions $\mathcal{L}$ are of form  $|\cdot|$  (line 223-229). Is this $l^2$ norm or $L^2$ norm or absolute value? Please specify.
>
> A2: The residual error in the convection-diffusion setup is defined as the mean square error on testing data points, which are generated on the mesh grid of the domain of interest $(x, t) \in [-10, \alpha] \times [0, 10]$ with $dx = 0.1$ and $dt = 0.1$. The $|\cdot|$ used in evaluating data and physics losses denote absolute values. We have added such information in Appendix C.4 to elaborate on evaluation metrics.
>
> Q3: How does a plot of number of NN parameters v.s. MSE. looks like? In table 2 only the relationship between number of control points v.s. MSE is reported.
>
> A3: We have added ablation experiments of NN size in Appendix D.5. In general we do not observe much difference in performance by varying the NN size, for our case study.
>
> Q4: Are all experiments conducted over noiseless setup? Is PI-DBSN robust against noisy observation? Is there any ablation study on how to choose $w_p$ and $w_d$ to attain better performance? In line 230, authors claim both are set to 1.
>
> A4: We have added ablation experiments on robustness against noise and different loss function weights in Appendix D.4. We observe that when the training data is noiseless, increasing data loss weight $w_d$ helps with performance as the network better learns the ground truth data. When the training data is noisy, increasing physics loss weight $w_p$ helps with enforcing the neighboring relationships in the PDE, resulting in better prediction accuracy.
>
> Q5: What is the performance of PI-DBSN over Burger's equation and Navier-Stokes equation using the same setup in FNO paper? Are there any specific reasons that the comparison ruled out FNO branch of method?
>
> A5: We have added experiment results on Burgers' equation in Appendix D.6. The results of PI-DBSN are comparable to FNO as reported in [1], in terms of prediction error.
>
> Q6: How is physics model loss being evaluated. Namely, what is the quadrature rule when computing $|\mathcal{F}_i(s, x, u)|^2$?
>
> A6: The physics model loss is evaluated as the squared residuals of the PDE on a mesh grid of the domain of interest, similar to PINNs [2]. We have elaborated the definition in the paper to $\mathcal{L}_p = \sum_i \sum _\mathcal{P} \frac{1}{|\mathcal{P}|} |\mathcal{F}_i(s, x, u)|^2$ where $\mathcal{P}$ is the set of points sampled to evaluate the governing physics model. Specifically, for the recovery probability experiment, $\mathcal{P}$ is the mesh grid on the domain of interest with $dx = dt = 0.1$, and for the 3D heat equation is the mesh grid with $dx = dt = 0.01$. We have added such information in the revised version of the paper.
>
> Q7: Though not required due to extensive work of experiments, I wish to, at least hear from authors, if complex geometric domain, such as $L$-shape problem introduced in Doleglo et al. (2022), or hyper-elastic problems setup in Geo-FNO can be computed via PI-DBSN.
>
> A7: Please see response to common questions in our General Response.
>
> [1] Li, Zongyi, et al. "Fourier neural operator for parametric partial differential equations." _arXiv preprint arXiv:2010.08895_ (2020).
>
> [2] Raissi, Maziar, Paris Perdikaris, and George E. Karniadakis. "Physics-informed neural networks: A deep learning framework for solving forward and inverse problems involving nonlinear partial differential equations." _Journal of Computational physics_ 378 (2019): 686-707.

---

> > ### Comment · Reviewer_dWHg · 2024-11-22
> >
> > Thank you for answering my questions. Two minor add-on questions:
> >
> > *    Why Figure 3 plot two different task losses altogether in one plot? Maybe change it to the pure residual loss plot will be more convincing.
> >
> > *    I appreciate the ablation of NN parameter complexity v.s. the performance. What I originally thought is: since the number of control points is different, the network output dimension is different. Therefore, I would like to know change of NN parameters versus number of control points with exact configuration listed in Table 2. Namely, can you add a line reflecting change of NN parameters in Table 2?

---

> > > ### Author Response · Authors · 2024-11-24
> > >
> > > Thank you for your constructive suggestions and clarifications on the questions. We have revised the paper accordingly to address your concerns.
> > >
> > > Q1: Why Figure 3 plot two different task losses altogether in one plot? Maybe change it to the pure residual loss plot will be more convincing.
> > >
> > > A1: We have updated Figure 3 to plot unnormalized physics loss plus data loss for clarity. We leave out ICBC losses for PINN and PI-DeepONet for fair comparisons. We also visualized the physics loss and data loss separately for all three methods in Appendix C.5. It is observed that PI-DBSN can achieve similar physics loss values compared with PINN, but converges faster. PI-DBSN can achieve lower data loss values than both baseline methods, likely due to its efficient representation of the solution space.
> > >
> > > Q2: Can you add a line reflecting change of NN parameters in Table 2?
> > >
> > > A2: We have added a line reflecting the number of NN parameters for different number of control points in Table 2. It can be seen that the number of NN parameters increases as the number of control points goes up. Analytically, the number of parameters in the last fully connected layer scales linearly with the total number of control points ($\ell_x \times \ell_t$ in the convection diffusion example), and the total number of NN parameters is this value plus the number of parameters in previous hidden layers.

---

> > > > ### Comment · Reviewer_dWHg · 2024-11-25
> > > >
> > > > Thank you for answering my questions and making an effort to revise the manuscript. I would raise my score to 6 as most of the concerns were addressed/resolved. The adaptation of PI-DBSN to complex geometry is important and I encourage author to continue working on this direction.

---

### Official Review · Reviewer_NS4y · 2024-11-03

**Soundness:** 3
**Presentation:** 2
**Contribution:** 2
**Rating:** 6
**Confidence:** 3

**Summary:**

This paper presents a novel Physics-Informed Deep B-Spline Network framework for efficiently solving partial differential equations (PDEs) with varying system parameters, initial conditions, and boundary conditions (ICBCs). Traditional physics-informed neural networks (PINNs) often struggle with discontinuous conditions and can be computationally demanding, as they typically require simultaneous learning of both weights and basis functions. The proposed method addresses these challenges by using a neural network to learn B-spline control points, which approximate the PDE solutions. This approach allows direct specification of ICBCs without imposing additional losses, reduces computational complexity by only requiring the learning of B-spline weights, and incorporates analytical formulas for calculating physics-informed loss functions. The paper further establishes theoretical guarantees that the B-spline networks serve as universal approximators for arbitrary-dimensional PDEs under specific conditions. Empirical results demonstrate that the B-spline network outperforms traditional methods, effectively handling discontinuous ICBCs and diverse initial conditions. This framework contributes a robust, efficient tool for solving high-dimensional PDEs, with applications in fields where complex physical systems require adaptive modeling.

**Strengths:**

The proposed Physics-Informed Deep B-Spline Network framework showcases originality in its formulation and application. By leveraging B-splines within a neural network architecture, the authors address a common challenge in physics-informed machine learning: the need for efficient, accurate solutions to partial differential equations (PDEs) with varying parameters and changing initial and boundary conditions (ICBCs). This approach creatively combines established techniques in both spline theory and neural networks to improve upon traditional physics-informed neural networks (PINNs), which typically require simultaneous learning of weights and basis functions. The novelty here also extends to how the B-spline networks are designed to handle discontinuous ICBCs—an area where many neural network approaches struggle. The method’s theoretical guarantee as a universal approximator further demonstrates its innovative edge, providing a rigorous mathematical foundation that strengthens its potential utility across a wide range of applications.

**Weaknesses:**

1. Limited Experiments
2. Scalability and Computational Complexity Analysis is not sufficient
3. Limited Discussion on Practical Constraints and Limitations
4. Lack of Comparison with Other Recent Advances

**Questions:**

1. Can you show more results on other PDEs, such as NS-Equation and so on?
2. Please compare with recent developed methods instead of original PINNs.

---

> ### Author Response · Authors · 2024-11-21
>
> Thank you for your comments.
>
> To address your concerns, we have added additional results on Burgers' equations (Appendix D.6), which is widely considered in the literature. Our proposed PI-DBSN method achieves comparable prediction results against Fourier Neural Networks (FNO) [1] in this test case. We have also added discussions on the limitations and future directions in the revised conclusion section. We point out that our proposed method has similar scalability compared with other physics-informed learning methods, but the computation complexity is drastically reduced because of the direct assignment of ICBC values and analytical calculation of derivatives. The advantageous training time is visualized in Figure 3 and quantitatively reported in Table 1.
>
> [1] Li, Zongyi, et al. "Fourier neural operator for parametric partial differential equations." _arXiv preprint arXiv:2010.08895_ (2020).

---

> > ### Comment · Reviewer_NS4y · 2024-11-26
> >
> > thank you for your response.

---

### Official Review · Reviewer_sH4h · 2024-11-04

**Soundness:** 2
**Presentation:** 3
**Contribution:** 2
**Rating:** 5
**Confidence:** 3

**Summary:**

In this paper, the authors present a hybrid framework which uses learning based B-spline model to approximate the PDEs solutions with different initial and boundary condtions. The proposed method only requires learning the weights of B-spline functions, which enables directly specifying IC/BC and calculating physics-informed loss through analytical formulas. It is demonstrated that the proposed method are universal appoxmiators of arbitrary dimensional PDEs on certain scenarios. The method outperforms the benchamrks on 3D heat equations.

**Strengths:**

The idea of using B-spline basis functions for PINN sounds novel. There is also a theoretical analysis to assisting the claim that PI-DBSN is a universal approximator under certain conditions. The advantages of analytically calculated gradients look sound, which can help stabilize and accelerate training process.

**Weaknesses:**

- In the experiments part, for 3D heat equation, it seems that there is no quantitative results for for the accuracy. It is unclear how the proposed method perform comparing to benchmarks especially PINN on this case.
- In Figure 3, though PI-DBSN converges faster than PINN, the final training loss is much higher than the averaged loss of PINN. In addition, the PI-DBSN almost converge in less than 500 epochs. Does the training process of PI-DBSN not converge properly? Or the test case is too simple so that the ability of the proposed method is not fully tested?
- The authors claim that "we can directly specify Dirichlet initial and boundary conditions through the control points tensor without imposing loss functions, which helps with learning extreme and complex ICBCs". It seems there is no test case with such a extreme and complex ICBCs in the paper. Therefore it is unclear how the proposed method shows advantages on ICBC. Would you minding providing such a case that benchmarks such as PINN and PI-DeepONet fail while the proposed method works?
- It seems that all the performance comparisons are all run on CPU, would you minding providing some results on GPU? If it is hard to get a GPU version, how do you think the results will diff from that on CPU? As GPU might be more friendly for methods with larger NNs for example PINN.
- What is the scale/problem size of the test cases? Can the proposed method scale to much larger test cases?
- The current test cases have relatively simple geometry i.e., rectangular domains, is it possible to apply the proposed method to scenarios with more complex boundary geometries (e.g., arbitrary geometry shape )? or non-convex domain (such as flow past a cylinder)

**Questions:**

See weakness.

---

> ### Author Response · Authors · 2024-11-21
>
> Thank you for your comments.
>
> Q1: In the experiments part, for 3D heat equation, it seems that there is no quantitative results for the accuracy. It is unclear how the proposed method perform comparing to benchmarks especially PINN on this case.
>
> A1: For the 3D heat equation case study, the average residuals for PI-DBSN during training and testing are $0.0028$ and $0.0032$, as reported in section 5.2. We have added the results of PINN on this example in Appendix C.6. The testing residual for PINN is $0.0121$, which is higher than the reported value ($0.0032$) for PI-DBSN.
>
> Q2: In Figure 3, though PI-DBSN converges faster than PINN, the final training loss is much higher than the averaged loss of PINN. In addition, the PI-DBSN almost converge in less than 500 epochs. Does the training process of PI-DBSN not converge properly? Or the test case is too simple so that the ability of the proposed method is not fully tested?
>
> A2: The loss functions of PINN and PI-DBSN are slightly different, as PINN takes initial condition and boundary condition (ICBC) losses as well, while PI-DBSN can directly specify ICBC values. The loss plot in Figure 3 is normalized, thus PINN appears to have lower overall loss values. From the prediction results we know that the test performance of PINN and PI-DBSN are comparable. However, the training of PI-DBSN is much faster than PINN - we do observe that PI-DBSN converges within ~500 epochs, thanks to the compact representation and the fact that only control points are learned.
>
> Q3: The authors claim that "we can directly specify Dirichlet initial and boundary conditions through the control points tensor without imposing loss functions, which helps with learning extreme and complex ICBCs". It seems there is no test case with such a extreme and complex ICBCs in the paper. Therefore it is unclear how the proposed method shows advantages on ICBC. Would you minding providing such a case that benchmarks such as PINN and PI-DeepONet fail while the proposed method works?
>
> A3: We believe the convection diffusion equation example in section 5.1 serves the purpose, as the initial and boundary condition is discontinuous at $(x, T) = (\alpha, 0)$. PI-DeepONet fails to learn this PDE, and PI-DBSN is much faster than PINN for this problem.
>
> Q4: It seems that all the performance comparisons are all run on CPU, would you minding providing some results on GPU? If it is hard to get a GPU version, how do you think the results will diff from that on CPU? As GPU might be more friendly for methods with larger NNs for example PINN.
>
> A4: We added experiments on GPU performance in the revised version of the paper (Appendix D.3). We observe consistent results compared to CPU performance (PI-DBSN faster than two baselines), while GPU will accelerate all methods as expected.
>
> Q5: What is the scale/problem size of the test cases? Can the proposed method scale to much larger test cases?
>
> A5: The scale/problem size of the proposed method is comparable to other physics-informed learning methods such as PINNs, where usually PDEs with dimensions less than 4 are considered. While the naive construction of the proposed method also suffers the curse of dimensionality, we do not see any issue if any problem-specific dimensionality reduction techniques can not be applied to our method if they could be applied to other physics-informed learning methods.
>
> Q6: The current test cases have relatively simple geometry i.e., rectangular domains, is it possible to apply the proposed method to scenarios with more complex boundary geometries (e.g., arbitrary geometry shape )? or non-convex domain (such as flow past a cylinder)
>
> A6: Please see responses to common questions in our General Response.

---

> > ### Comment · Reviewer_sH4h · 2024-11-26
> > **response**
> >
> > Thanks for the reply. Most of my concerns are resolved.

---

### Author Response · Authors · 2024-11-21
**General Response**

We sincerely thank all the reviewers for their insightful comments and constructive feedback. We have revised our paper accordingly, and below is the summary of changes and the responses to common questions.

**Summary of Revision**

1. We added details of the evaluation metrics used in the paper in Appendix C.4.
2. We reported experiment results on GPU in Appendix D.3 and reported PINN performance on 3D heat equations in Appendix C.6 to address Reviewer sH4h’s concerns.
3. We conducted ablation experiments on the robustness against noise and different configurations of loss function weights, and reported our results in Appendix D.4.
4. We conducted ablation experiments on NN size and number of parameters, and reported the results in Appendix D.5.
5. We conducted additional experiments on Burgers’ equations and presented the results in Appendix D.6. The prediction error is comparable to Fourier Neural Operators as reported in the original paper.

**Response to Common Questions**

Q1: Are there any comparisons to several advanced neural operator methods?

A1: We hope to point out that the advantage of the proposed method lies in the better speed and accuracy tradeoffs, instead of accuracy alone. Neural operator methods such as Fourier Neural Operators (FNOs) do achieve good accuracy on complex problems such as Naiver Stokes equations, but usually require larger neural networks for representation and take long times to train. On the other hand, our method is more lightweight and achieves low prediction error within a very short training time, thanks to the compact representation of B-splines, direct assignment of ICBCs, and analytical calculation of derivatives. We expect the practical application cases of the proposed method to be moderate problems with contingent computation limitations (e.g. edge devices).

Q2: The current test cases have relatively simple geometry i.e., rectangular domains, is it possible to apply the proposed method to scenarios with more complex geometries? If so, how the spline basis representation might need to be adapted?

A2: The current setting of the method can be directly applied to complex geometries, but will lose the direct assignment of ICBC value property (the analytical derivative property remains). This is because one can always find a rectangular-like super-set of the domain of interest, and train a PI-DBSN to learn the solution values on that domain. Then the ICBC losses need to be imposed during training, similar to PINNs.

If one wants to preserve the direct assignment of ICBC values property, the following two approaches can be possibly studied.

1. If the desired domain is diffeomorphic to the original domain, one can add another transformation after the B-spline prediction module of the framework to map to the desired domain. Due to the diffeomorphism, one has the forward and inverse mapping of the values and derivatives of the solution surfaces, which enables forward and backward passes of data loss and physics loss through the additional transformation to the proposed PI-DBSN.

2. The other option is to use Non-Uniform Rational B-Splines (NURBS) [1], which are more generic B-splines that can approximate complex high-dimensional geometries. The derivatives have analytical formulas as well, but are different and more complicated than standard B-splines.

Both approaches require additional theoretical and empirical analysis, but we believe they are exciting future directions.

We have updated the conclusion section of the paper to reflect those discussions.

[1] Piegl, Les, and Wayne Tiller. _The NURBS book_. Springer Science & Business Media, 2012.

---

### Meta-Review · Area_Chair_2oyH · 2024-12-21

**Metareview:**

The paper introduces physics-informed deep B-spline networks (PI-DBSNs). The high level idea is to start with a physics-informed neural network (PINN), which outputs the solution of a PDE at each point, and instead using a change of basis, train the network to output coefficients+control points of the B-spline representation of the solution. This approach has many benefits; the authors theoretically explore some of these benefits, and also show improvements on simple systems over competing methods such as PINNs and DeepONets.

The paper is generally well-written, the main idea makes sense, and the paper is a relatively self-contained complete contribution. In particular I appreciated the fact that the authors explore reasonably complex initial/boundary conditions.

However, my own opinion is a bit more critical than those of the reviewers. The biggest weaknesses of the paper are:
(a) novelty
(b) weak baselines
(c) only applicable to simple geometries.
See the point on novelty below.
The comparisons are with PINNs and DeepONets --- which are influential papers --- but hundreds of follow-up methods have been published on this topic before and I would have expected a more robust suite of comparisons on standard benchmarks (such as PDE-Bench). The point with weak baselines [1] is a particular problem in the scientific ML community, and more stringent acceptance criteria may be one way to solve it. Finally, the factorized approximation for higher dimensional splines limits the method to be applicable only to simple rectangular geometries.


[1] "Weak baselines and reporting biases lead to overoptimism in machine learning for fluid-related partial differential equations", McGreivy and Hakim, Nat. Mach. Intelligence, 2024.

**Additional Comments On Reviewer Discussion:**

A couple of reviewers pointed out missing references and called out a lack of comparisons with previous work. In particular, a reviewer pointed out the existence of several papers which do combine splines with PINNs, so the novelty of the basic idea is a but diminished. The authors gave a reasonable response to this question, but let me also point out another missing reference [2], which ialso integrates B-splines and NURBS within PINNs.

[2] "Differentiable Spline Approximations", Cho et al, NeurIPS 2021.

---

### Decision · Program_Chairs · 2025-01-22

Reject